# Severe anaemia complicating HIV in Malawi; Multiple co-existing aetiologies are associated with high mortality

Minke H. W. Huibers[1,2]*, Imelda Bates[3], Steve McKew[3,4], Theresa J. Allain[5], Sarah E. Coupland[6,7], Chimota Phiri[5], Kamija S. Phiri[8], Michael Boele van Hensbroek[1], Job C. Calis[1,9,10]

1 Global child health group, Emma Children's Hospital, University Medical Centres Amsterdam, location Academic Medical Centre, University of Amsterdam, The Netherlands, 2 Amsterdam Institute of Global Health Development, Amsterdam, the Netherlands, 3 Liverpool School of Tropical Medicine, Liverpool, United Kingdom, 4 Department of Internal Medicine, Shrewsbury and Telford Hospital NHS Trust, Shrewsbury, United Kingdom, 5 Department of Internal Medicine, College of Medicine, Queen Elizabeth Central Hospital, Blantyre, Malawi, 6 Department of Molecular and Clinical Cancer Medicine, Institute of Translational Medicine, University of Liverpool, Liverpool, United Kingdom, 7 Department of Pathology, Royal Liverpool University Hospital, Liverpool, United Kingdom, 8 School of Public Health and Family Medicine, College of Medicine, Blantyre, Malawi, 9 Department of Pediatric Intensive Care, Emma Children's Hospital, Academic Medical Centre, University of Amsterdam, The Netherlands, 10 Department of Paediatrics, College of Medicine, Queen Elizabeth Central Hospital, Blantyre, Malawi

* mhw.huibers@gmail.com

**Data Availability Statement:** All relevant data are within the manuscript and its Supporting Information files.

## Abstract

### Background

Severe anaemia is a major cause of morbidity and mortality in HIV-infected adults living in resource-limited countries. Comprehensive data on the aetiology are lacking but are needed to improve outcomes.

### Methods

HIV-infected adults with severe (haemoglobin ≤70g/l) or very severe anaemia (haemoglobin ≤ 50 g/l) were recruited at Queen Elizabeth Central Hospital, Blantyre, Malawi. Fifteen potential causes and associations with anaemia severity and mortality were explored.

### Results

199 patients were enrolled: 42.2% had very severe anaemia and 45.7% were on ART. More than two potential causes for anaemia were present in 94% of the patients including iron deficiency (55.3%), underweight (BMI<20: 49.7%), TB infection (41.2%) and unsuppressed HIV infection (viral load >1000 copies/ml) (73.9%). EBV/CMV co-infection (16.5%) was associated with very severe anaemia (OR 2.8 95% CI 1.1–6.9). Overall mortality was high (53%; 100/199) with a median time to death of 17.5 days (IQR 6–55) days. Death was associated with folate deficiency (HR 2.2; 95% CI 1.2–3.8) and end stage renal disease (HR 3.2; 95% CI 1.6–6.2).

**Funding:** Dr. Steve McKew received a grant from the Wellcome Trust fund; Project number: WT086559, Liverpool, United Kingdom grant. Data collection and laboratory testing were all paid by this grant. Drs. M. Huibers received a grant from the Nutricia research foundation; Project number 2017-43 The Hague, The Netherlands. All other authors were funded by there own institutions. The funders had no role in study design, data collection and analysis, decision to publish, or preparation of the manuscript.

**Competing interests:** The authors have declared that no competing interests exist.

## Conclusion

Mortality among severely anaemic HIV-infected adults is strikingly high. Clinicians should be aware of the urgent need for a multifactorial approach including starting or optimising HIV treatment, considering TB treatment, nutritional support and optimising renal management.

## Introduction

Anaemia is recognized as the most common haematological complication of Human Immunodeficiency Virus (HIV) infection worldwide [1,2] The World Health Organization (WHO) defines anaemia as a haemoglobin level below 110–120 g/l. In sub-Saharan Africa, 60% of HIV-infected adults are anaemic and 22% are severely anaemic [3,4]. Anaemia is associated with an increased one-year mortality in HIV infection of 8%, which rises to 55% in those with severe anaemia [5,6]. Anaemia treatment may even improve survival [7]

To prevent and treat severe anaemia in HIV-infected patients, a comprehensive understanding of the aetiology and pathophysiology is essential. Severe anaemia in HIV infection has been associated with micronutrient deficiencies, infections (viral, bacterial and parasitic) and inflammation, medication induced (zidovudine and cotrimoxazole) and neoplastic diseases [8–12]. Only a few studies have comprehensively studied the multifactorial aetiology and pathophysiology of HIV-associated severe anaemia in sub-Saharan Africa despite the high burden of HIV infection in this region [2]. Commonly, studies only report on the association between HIV infection and a single cause of severe anaemia, for example iron deficiency, without considering the multiple causes of severe anaemia that may impact on an HIV-infected patient [13,14]. As a consequence, evidence to inform preventive or treatment guidelines for severe anaemia in HIV-infected patients in sub-Saharan Africa is scarce. In practice, severe anaemia management in HIV-infected patients is often still based on the same strategies used for non-HIV infected patients, including iron supplementation, malaria treatment and deworming [2,15]. These strategies may be ineffective as the causes of HIV-associated severe anaemia may be different, and even harmful, since iron supplementation may exacerbate infections and potentially cause deterioration of the patient's condition [16,17].

Better knowledge about the aetiology of severe anaemia is essential in order to develop evidence-based protocols and ultimately improve outcomes for HIV-infected adults with severe anaemia in sub-Saharan Africa. To address this knowledge gap, we performed a comprehensive observational cohort study to explore the prevalence of potential aetiologies of HIV-associated severe anaemia in Malawian in-patients, and studied associations between these and the severity of the anaemia and patient outcomes.

## Methods

An observational cohort study of HIV-infected patients with severe anaemia (haemoglobin ≤70 g/l) admitted to the Queen Elizabeth Central Hospital (QECH), Blantyre, Malawi between February 2010 and March 2011 was performed. All HIV-infected patients above 18 years of age with severe anaemia admitted to the general medical ward were approached and enrolled in the study if they provided informed consent. A case record form including a detailed medical history and physical examination was completed for each enrolled patient. On admission a venous blood sample was collected and a chest X-ray was performed. The patients were managed according to the hospital protocols, which included a blood transfusion if required,

treatment of correctable conditions including anti-malarial medication and antibiotics. Anti-retroviral treatment (ART) was provided according to the national Malawi guidelines, which stipulated that ART should only be initiated by specialist outpatient clinicians, so patients were not started on ART on in-patient wards. At the time of the study first line ART was a combination of stavudine, lamivudine and nevirapine and second line treatment was a combination of zidovudine, lamivudine, tenofovir and lopinavir/ritonavir [18]. ART was only prescribed for WHO stage 3 and 4 disease, or WHO stage 1 and 2 disease with CD4 count $< 350 \times 10^9$/l [18,19]. Cotrimoxazole was prescribed routinely to all HIV-infected patients on ART as *Pneumocystis Jirovecii* prophylaxis. For this study patients were followed up in a dedicated ART clinic after discharge. Follow-up was done for a maximum of 365 days after enrolment, or when they attended the ART clinic for routine appointments or, if they failed to attend, by a home visit from a study nurse.

## Laboratory assays

All samples were analysed within 24 hours of collection or stored at -80˚C for further analysis. On enrolment haemoglobin concentrations were measured on the ward using the HemoCue B-Haemoglobin analyser (HemoCue, Ängelholm, Sweden) to screen for eligibility. For patients enrolled in the study, the haemoglobin and red cell indices (MCV, MCH and MCHC) were determined using an automated analyser (Beckman Coulter, Durban, South Africa). CD4-cell counts were assessed using BD FACS Count (BD Biosciences, San Jose, CA, USA). Transferrin, iron, ferritin, folate and vitamin B12 were analysed on Modular P800 and Monular Analytic E170 systems (Roche Diagnostics, Switzerland). Soluble transferrin receptor (sTfR) levels were measured using ELISA (Ramco Laboratories, TX, USA). Serum creatinine was analysed using a Beckman Coulter CX5 (ADVIA 2400 Siemens Healthcare Diagnostics). Renal function was measured by estimating glomerular filtration rate (eGFR) using simplified Modification of Diet in Renal Disease (MDRD)-Study formula and the GFR was classified by Chronic Kidney Disease classification [20–22]. For all tests the manufacturers' reference ranges were used; internationally accepted cut-offs were used to define deficiencies [23]. Thick blood films were prepared and stained for malaria microscopy. Malaria was defined as the presence of *Plasmodium falciparum* asexual parasites in the blood films. HIV infection was confirmed using two point-of-care antibody tests (Unigold® and Determine®). For blood cultures a venous blood sample was inoculated into BACTEC Myco/F-Lytic culture vials and incubated in a BACTEC 9050 automated culture system (Becton Dickinson) for 56 days. Sub-culturing blood and sputum, susceptibility testing and isolate identification were performed by standard techniques [23,24]. Likely contaminants were recorded as absence of pathogens. Sputum cultures were examined for mycobacteria using Ziehl–Nielsen staining. Whole-blood isolates were assessed for Epstein–Barr virus and cytomegalovirus infection by semi-quantitative PCR and for parvovirus B19 by real-time PCR [25]. All chest X-rays were reviewed by a radiologist for signs of pulmonary tuberculosis (TB). When TB was suspected, standardized treatment was started according to the local protocols.

## Bone marrow

If the patient's clinical condition allowed and they provided consent, a bone marrow aspirate and trephine biopsy were performed. All bone marrow samples were taken from the posterior iliac crest. Samples of the aspirates were spread onto slides and trephine biopsies were fixed, decalcified and embedded in paraffin wax [26, 27]. Bone marrow samples were sent to the Haematopathology Referral Centre at the Royal Liverpool University Hospital, Liverpool UK, for analysis. Sections of the trephine blocks were stained with haematoxylin and eosin, and

Giemsa, with Perls stain for iron, and for reticulin [26]. Bone marrow analysis was performed by pathologists who were unaware of the patients' data. All slides were examined for using a predefined format and diagnoses were allocated to the categories lymphoproliferative disease, myeloproliferative disease, myelodysplastic syndrome (MDS), TB and 'other' [28–30]. The need for additional histochemical (e.g. Ziehl-Neelsen) or immunohistological (e.g. CD3, CD20) staining was determined according to the local protocol in Liverpool depending on the preliminary morphological findings.

## Definitions for potential factors involved in the aetiology of severe anaemia

A total of 15 potential factors involved in the aetiology of severe anaemia were investigated. Factors were based on two previous studies performed in Malawi and a systemic review on this topic [31–33].and comprised (with definitions) 1) Unsuppressed HIV-infection; viral load ≥1000 copies/ml. 2) TB: presence of one or more of the following: a) positive sputum culture, b) chest X-ray with signs of pulmonary TB and/or c) on going TB treatment at time of enrolment d) clinical diagnosis by local doctor including unknown generalized lymphadenopathy and/or night sweats of > 30 days and of unknown origin e) caseating granulomata in the bone marrow trephine. 3) Malaria: presence of malaria parasites in a thick blood film. 4) parvovirus B19: viral load of >1000 copies/ml. 5) Cytomegalovirus (CMV); viral load of >100 copies/ml. 6) Epstein-Barr virus (EBV); viral load of >100 copies/ml. 7) Bacteraemia; a blood culture growing a potential pathogen. 8) Underweight (BMI ≤18.5). 9) Serum folate deficiency (≤3 ng/l). 10) Vitamin B12 deficiency (≤180 pg/ml). 11). Iron deficiency; In a sub study (Huibers et al PLOS One 2019) we evaluated bone marrow (BM) iron deficiency using several conventional blood markers; MCV (fl), MCH (pg/cells), Fe (umol/l), ferritin (ug/dl), TFr1 receptor (nmol/l), TrF index (stFR/Log ferritin). All markers showed suboptimal correlations (i.e. $AUC^{ROC} < 0.6$) with BM iron deficiency, though MCV performed best ($AUC^{ROC}$ 0.545). MCV was therefore used to identified iron deficiency as it was the best (though suboptimal) conventional marker in our setting. It was also available for patients beyond the subgroup (n = 76) who had a bone marrow result and it is available in most African settings [3, 22, 33]. Since an MCV <83fL is commonly used in other studies and guidelines [3, 22, 33] and it was the best predictor of BM iron deficiency in our setting, we used it as the marker for iron deficiency in this analysis. As the AUC-ROC was suboptimal we chose to call it MCV ≤83fL rather than iron deficiency. 12) Zidovudine usage. 13) Cotrimoxazole usage. 14) Bone marrow disorders; lymphoproliferative disease, myeloproliferative disease or MDS. 15) Renal impairment: a GFR which either indicated impaired (GFR 15–59 ml/min/1.73 $m^2$) or End Stage (GFR ≤15 ml/min/1.73 $m^2$) Renal Disease [22,35]].

## Statistics

The study was primarily designed to give a complete overview of the potential factors associated with severe anaemia, including lymphomas, in severely anaemic African patients infected with HIV. The selected sample size of 200 would be able to detect an estimated prevalence of 5% or 10% with confidence intervals of 2.4% - 9% and 6–15% respectively. Baseline characteristics and prevalence of potential risk factors are presented as proportions or medians with IQR. Logistic regression was performed to model the association between anaemia and potential factors associated with anaemia. Results are expressed as OR with 95% CI and p-values. Variables associated with the outcome variables ($P \leq 0.10$) in the univariate analysis were included in the multivariate model in a stepwise approach. Kaplan Meier survival curves were used to assess cumulative mortality. Significant differences were investigated with a Log Rank test. Uni- and multivariate analyses were done using logistic regression and Cox regression to

describe predictors of overall mortality. A sub- analysis was performed for mortality within 60 days. A sub analysis was performed to evaluate the distribution of co-existing factors between mortality within and after 60 days. Group comparisons for categorical data were performed using the χ2 test or Fisher's exact test, and for continuous data using the *t*-test or the Wilcoxon rank-sum test. P values of < 0.05 were regarded as statistically significant. All reported P values were two-sided. The data were analysed using Stata (version 12) (STATA Corp. LP, Texas, TX, USA).

## Ethics

The Research Ethics Committee of the College of Medicine, University of Malawi (P.09.09.824) and the Research Ethics Committee of Liverpool School of Tropical Medicine (research protocol 09.64) approved the study. The purpose of the study was explained to the patients in the local language (Chichewa) and written informed consent was obtained before inclusion in the study.

## Results

In total, 199 patients were included in the study: 64.8% were female. The median age was 32 years (IQR 27–61 years). The median haemoglobin was 53 g/l (IQR 4.2–6.3) and 84 (42%) patients had very severe anaemia (Hb ≤ 50 g/l). A total of 91 (45.7%) patients were on ART at enrolment including 79.1% on first line ART. During the study period, an additional 41 (21%) patients started on ART treatment. 67.1% of the patients were immune suppressed with a CD4 count ≤ 200 cells/mm. Baseline characteristics of the patients are shown in Table 1.

The prevalence of factors and their association with severe anaemia is shown in Table 2. An unsuppressed HIV-infection occurred in 73.9% patients. TB was the second most common infection occurring in 82 (41.2%) of the patients. In 19 (23%) of these patients TB was diagnosed on their chest X-ray. Granulomata were seen in the bone marrow trephine in 15 (18%) of all the 82 patients that had a diagnosis of TB. 11 (13%) patients were on TB treatment at enrolment. 69/170 (40.5%) patients had evidence of current EBV infection and 57/170 (33.5%) had evidence of current CMV infection. Co-infection with CMV and EBV was found in 28/170 (16.5%) of the patients. Bacteraemia was diagnosed on a positive blood culture in 26 (13.1%); the most common pathogens were E. Coli (12 patients; 42.9%) and non-Typhoid Salmonella (5 patients; 19.9%). 74/148 (49.7%) patients were underweight and MCV ≤ 83 fl occurred in 61/180 (33.9%) of the patients. Bone marrow sampling was performed in 73 patients. Of these, 28 (38.4%) had morphological abnormalities with MDS being the most common abnormality (20 patients; 27.4%) (Table 2). Renal impairment was diagnosed in 36/185 patients (19.5%) and 12 of these patients (33%) had end stage renal disease. Overall, patients had a mean of 3 (range 1–8) co-existing aetiologies of severe anaemia (Fig 1). The overlap of factors is visually displayed in supplemental S1 Fig.

Comparing the different risk factors between very severe anaemia (Hb < 50g/L) and severe anaemia, EBV/CMV co-infection (OR 2.8 95% CI 1.1–6.9) was the only factor associated with very severe anaemia (Table 2).

During the one-year follow-up period, 101 study patients (50.8%) died. The median time to death was 17.5 days (IQR 6–55) and 81 (80.2%) of these deaths occurred within 60 days of admission (Fig 2). The factors associated with mortality within 60 days and those associated with overall mortality in HIV-infected adults in Malawi are provided in S1 Table. Folate deficiency and end stage renal disease were associated with overall mortality with Hazard Ratio 2.0 (95% CI 1.2–3.6) and Hazard Ratio 3.0 (95% CI 1.5–5.9) respectively (Fig 3). Only end stage renal disease (Hazard Ratio 2.7 95% CI 1.2–6.2) was associated with mortality within 60 days

**Table 1. Baseline characteristics of HIV-infected patients with severe anaemia at enrolment into study.**

| | Overall |
|---|---|
| Age, years (median IQR) | 32 (IQR 27–61) |
| Sex (female) | 129/199 (64.8%) |
| **Haematology** | |
| Haemoglobin (Hb) (median, IQR) g/l | 53 (IQR 42–63) |
| Severe anaemia (Hb 51–70 g/l) | 115/199 (57.8%) |
| Very severe anaemia (Hb ≤ 50g/l) | 84/199 (42.2%) |
| Pancytopenia[1] | 42 /184 (21.2%) |
| **Mortality** | |
| Overall mortality (365 days) | 101/199 (50.8%) |
| Early mortality (60 days) | 81/101 (80.2%) |
| Days until death (median, IQR) | 17.5 days (6–55) |
| **HIV-disease and treatment** | |
| CD4 (median, IQR) | 175 (IQR 55–825) |
| CD4 ≤200 cells/mm$^3$ | 104/155 (67.1%) |
| Viral load ≥1000 copies/ml | 136/184 (73.9%) |
| ART at enrolment | 91/199 (45.7%) |
| First-line ART | 72/199 (36.2%) |
| Second-line ART | 11/199 (5.5%) |
| Non-specified ART | 8/199 (4%) |
| **Blood transfusions and supplemental treatment** | |
| Blood transfusion at enrolment | 47/199 (23.6%) |
| Folate supplementation at enrolment | 57/199 (28.6%) |
| Iron supplementation at enrolment | 81/199 (40.7%) |
| Vitamin B12 supplementation at enrolment | 0/199 (-) |

[1] Pancytopenia is defined as thrombocytopenia (≤150 x 10$^9$/l) and leucopenia (≤4 x 10$^9$/l) and severe anaemia (Hb ≤ 70 g/l) [22]. Abbreviations: Hb: Haemoglobin, ART: antiretroviral therapy.

(S2 Table). Neither very severe anaemia (haemoglobin ≤50 g/l) nor the haemoglobin levels were associated with mortality in the study patients (Hazard Ratio 0.9, 95%CI 0.6–1.4 and Hazard Ratio 1.01, 95% CI 0.9–1.2 respectively).

## Discussion

In this study we described the prevalence of several potential aetiologies for severe and very severe anaemia in HIV-infected Malawian adults. Patients had a mean of three co-existing aetiologies potentially contributing to their anaemia, the most common one being unsuppressed HIV infection. Mortality in the study patients was extremely high, as 51% of the patients died within one year and most died within 60 days of admission. Severe anaemia in HIV-infected patients in a resource limited setting, such as Malawi, is therefore a multi-causal critical condition associated with high mortality.

Anaemia in HIV-infected patients is an independent predictor of mortality with deaths increasing as haemoglobin concentrations decrease [6, 35]. However HIV pathogenesis is complex and although anaemia may contribute to morbidity and mortality, the data suggest that the aetiology is multi-causal. Previous studies have reported an estimated one-year mortality of 30–55% in severely anaemic, compared to 3.7% in non-anaemic, HIV-infected patients in resource limited settings [5, 6]. Our study results are consistent with these outcomes. The seriousness of severe anaemia and its complexity in HIV-infected patients needs to be better recognized. Whether severe anaemia is the direct cause of the mortality or indirectly

**Table 2. Distribution and multivariate analysis of co-existing factors associated with severe (Hb≤ 70 g/l- Hb> 50 g/l) and very severe anaemia (Hb≤ 50 g/l) in HIV-infected adults in Malawi.**

| | Overall N = 199 (100%) | Severe anaemia N = 115/199 (57.8%) | Very severe anaemia N = 84/199 (42.2%) | Univariate | | | Multivariate | | |
|---|---|---|---|---|---|---|---|---|---|
| | | | | Odds | 95% -CI | P-value | Odds | 95% -CI | P-value |
| Sex (female) | 129 (64.8%) | 74/115 (65.2%) | 55/84 (65.4%) | 1.1 | 0.6–1.9 | 0.869 | - | - | - |
| **HIV** | | | | | | | | | |
| CD4 ≤200 cells/mm³ | 104/155 (67.1%) | 62/87 (71.3%) | 42/68 (61.8%) | 1.04 | 0.6–1.8 | 0.889 | - | - | - |
| Viral load ≥1000 copies/ml | 136/184 (73.9%) | 81/104 (77.9%) | 55/80 (68.8%) | 0.6 | 0.3–1.2 | 0.163 | - | - | - |
| On ART at enrolment | 91/199 (42.7%) | 52/115 (40.9%) | 39/84 (45.2%) | 1.1 | 0.6–1.8 | 0.865 | - | - | - |
| **Infection** | | | | | | | | | |
| Malaria | 6/167 (1%) | 3/100 (3.0%) | 3/67 (4.5%) | 1.5 | 0.3–7.8 | 0.617 | - | - | - |
| Tuberculosis | 82/199 (41.2%) | 53/115 (46.0%) | 29/84 (34.5%) | 0.7 | 0.6–0.99 | 0.043 | 0.6 | 0.1–2.8 | 0.507 |
| Bacteraemia[1] | 26/199 (13.1%) | 17/115 (61.7%) | 9/82 (11.0%) | 0.7 | 0.3–1.6 | 0.402 | - | - | - |
| Parvovirus B19 | 7/170 (4.2%) | 5/99 (5.0%) | 2/71 (2.8%) | 0.5 | 0.1–2.9 | 0.476 | - | | |
| Cytomegalovirus (CMV) | 57/170 (33.5%) | 28/99 (28.3%) | 29/71 (40.8%) | 1.8 | 0.9–3.3 | 0.088 | - | - | |
| Epstein-Barr virus (EBV) | 69/170 (40.6%) | 40/99 (40.4%) | 29/71 (40.8%) | 1.0 | 0.6–1.9 | 0.954 | - | - | - |
| EBV/CMV co-infection | 28/170 (16.5%) | 12/99 (12.1%) | 16/71 (22.5%) | 2.1 | 0.9–4.5 | 0.075 | 2.8 | 1.2–7.0 | 0.024 |
| **Malnutrition** | | | | | | | | | |
| Underweight | 74/148 (49.7%) | 40/81 (49.3%) | 34/68 (50.0%) | 1.0 | 0.5–2.0 | 0.940 | - | - | - |
| Vitamin B12 deficiency | 2/194 (1.0%) | 1/113 (0.8%) | 1/81 (1.2%) | 1.4 | 0.09–22.7 | 0.813 | - | - | - |
| MCV ≤ 83 fl[2] | 61/180 (33.9%) | 38/104 (36.5%) | 23/76 (30.3%) | 0.8 | 0.40–1.42 | 0.380 | - | - | - |
| Folate deficiency | 23/194 (11.9%) | 13/113 (11.5%) | 10/84 (11.9%) | 1.1 | 0.5–2.6 | 0.858 | - | - | - |
| **Medication** | | | | | | | | | |
| Cotrimoxazole | 163/199 (81.9%) | 99/115 (81.6%) | 64/84 (76.2%) | 0.5 | 0.3–2.3 | 0.076 | 0.5 | 0.2–1.2 | 0.120 |
| Zidovudine | 13/199 (6.5%) | 5/50 (10.0%) | 8/40 (20.0%) | 2.3 | 0.7–7.5 | 0.187 | - | - | - |
| **Renal function** | | | | | | | | | |
| Impaired (GFR 15–60) | 24/185 (13.0%) | 12/105 (11.4%) | 12/80 (15.0%) | 1.5 | 0.6–3.6 | 0.339 | | | |
| End stage (GFR ≤15) | 12/185 (6.5%) | 3/105 (2.9%) | 9/80 (11.3%) | 4.6 | 1.2–17.6 | 0.027 | 4.0 | 0.9–16.8 | 0.061 |
| **Bone marrow** | | | | | | | | | |
| Bone marrow disease | 28/71 (39.4%) | 19/42 (45.2%) | 9/29 (31.0%) | 0.5 | 0.2–1.5 | 0.231 | - | - | |
| **Aetiology** | | | | | | | | | |
| Co-existing aetiologies per patient (mean, SD) | 3.3 (1.3) | 3.3 (1.2) | 3.2 (1.4) | 0.8 | 0.4–1.8 | 0.605 | - | - | - |

[1] A total of 28-blood cultures were positive, the most common organisms were *E. coli* (42.9%; 12/28) and non-Typhoid *Salmonella* (17.9%; 5/28). [2] Iron deficiency was defined by MCV≤ 83 fl. Explanatory variables associated with the outcome variables (P > 0.10) in the univariable analysis were excluded in the multivariable model in a stepwise approach (-). Abbreviation: GFR; Glomerular filtration rate.

responsible through other contributing factors, severe anaemia should be recognized by clinicians as a sign of critical HIV disease and should therefore should be classified as a stage 4 condition instead of its current stage 3 classification [36, 37].

Unsuppressed HIV virus was present in 79% of the patients in our study population. HIV may cause anaemia directly through an inhibitory effect of the HIV-virus on the erythropoietin progenitor cells in the bone marrow, or indirectly through opportunistic infections and/or inflammation causing anaemia [38]. A large cross-sectional study in Tanzania showed that the risk of developing severe anaemia in HIV-infected patients was increased two- to three-fold among patients with advanced HIV disease [3]. Our findings, and those of previous studies,

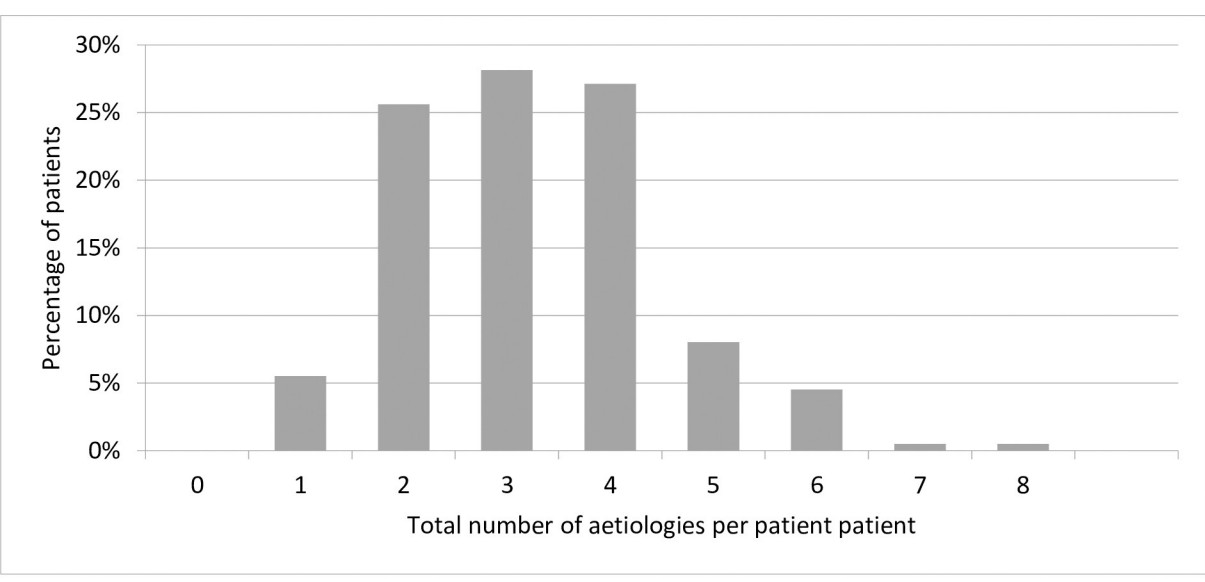

**Fig 1. Total number of aetiologies for severe anaemia co-existing in each patient (n = 199).** Mean is 3 factors (SD 1.3), range 1–8. Aetiologies for severe anaemia include: 1) Unsuppressed HIV-infection; viral load ≥1000 copies/ml. 2) TB: one or more of the following were present: a) positive sputum culture, b) chest X-ray with signs of pulmonary TB and/or c) on going TB treatment at time of enrolment d) clinical diagnosis by local doctor including unknown generalized lymphadenopathy and/or night sweats of > 30 days and of unknown origin e) caseating granulomata in the bone marrow trephine. 3) Malaria: presence of malaria parasites in a thick blood film. 4) Parvovirus B19: viral load of >1000 copies/ml. 5) Cytomegalovirus (CMV); load of >100 copies/ml. 6) Epstein-Barr virus (EBV); viral load >100 copies/ml. 7) Bacteraemia; a blood culture growing a potential pathogen. 8) Underweight (BMI ≤18.5). 9) Serum folate deficiency (≤3 ng/l). 10) Vitamin B12 deficiency (≤180 pg/ml). 11). Iron deficiency defined by MCV ≤ 83 fl. 12) Zidovudine usage. 13) Cotrimoxazole usage. 14) Bone marrow disorders; lympho-proliferative disease, myeloid-proliferative disease or MDS. 15) Renal impairment: a GFR which either indicated impaired (GFR 15–59 ml/min/1.73 m$^2$) or End Stage (GFR ≤15 ml/min/1.73 m$^2$) Renal Disease [22, 35].

indicate that controlling HIV infection by starting or switching ART treatment should be prioritised as the most important and urgent step in treatment protocols for severely anaemic HIV-infected patients. After our study had been completed guidelines for initiating ART changed. At the time of the study the trigger for starting ART was based on a patient's CD4 count, whereas it is currently recommended that ART should be started early in the course of HIV disease [18, 37]. It will be important for the impact of this policy change on HIV-related anaemia, and its consequences, to be evaluated. Irrespective of the policy change for initiating ART, the findings of our study remain very relevant because many HIV-infected patients in resource-limited settings present late in the course of their disease or are unable to access reliable supplies of ART. Consequently this patient population is likely to experience high levels of life-threatening anaemia since it is closely associated with severe HIV disease and may be exacerbated by the effect of medication such as zidovidine and cotrimoxazole on the bone marrow [39, 40].

TB has previously been associated with anaemia in HIV-infected patients [40]. In our study patients TB was a common co-infection with HIV (41%). This prevalence is comparable to previous reports on TB prevalence (43%) among severely anaemic HIV-infected patients in Africa [9, 15]. The pathophysiology of TB-associated anaemia in HIV-infected patients remains unclear. Bone marrow invasion by TB organisms or altered iron metabolism, as a side effect of tuberculostatic drugs, have been described [41]. Only 13% of patients were on TB treatment at enrolment. TB medication itself was not associated with the severity of anaemia (OR 1.6 95%CI 0.7–3.8). Given that we and others have found TB in nearly half of the HIV-infected patients with severe anaemia, TB screening and rapid initiation of treatment should

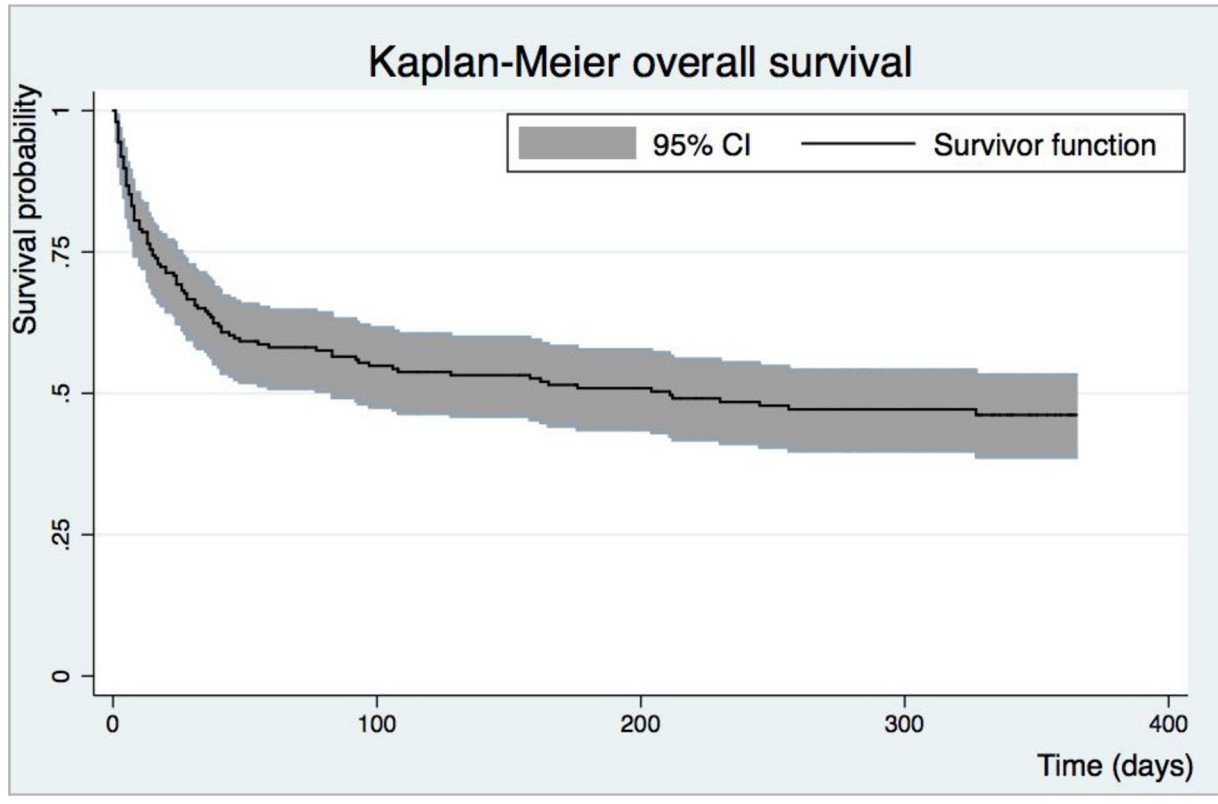

**Fig 2. Kaplan Meyer survival curve over time (days) for adult Malawian patients with HIV infection and severe anaemia during 365 days follow-up.** Abbreviation: 95% confidence interval (95% CI).

be a high priority in the management of these patients, especially in resource limiting settings where there is a high TB prevalence.

Viral infections such as EBV, CMV and parvovirus B19 have been associated with anaemia in HIV-infected patients [42, 43]. In our study EBV and CMV were common and present in 40% and 35% of patients respectively, whilst parvovirus B19 was less common (4%). Although parvovirus B19 is pathophysiologically linked to mild anaemia, its role in the development severe anaemia in HIV infected patients has not been clearly established [43, 44]. Data on co-infections are very limited in African patients [33, 45, 46]. Our study is the first to describe the association between co-existing CMV and EBV infections and very severe anaemia (haemoglobin ≤50 g/l] which may be related to direct viral inhibition of erythropoiesis [45, 47, 48]. The majority of our patients with co-infection had advanced HIV disease (88.5%) but the association between EBV and CMV co-infection and very severe anaemia remained significant even after correction for advanced HIV disease. As CMV is a treatable infection it will be important to determine the effect of ART on CMV infection and severe anaemia, as ART can reduce CMV infection by improving immune status [49].

Malaria is unsurprisingly associated with anaemia in HIV-infected patients in sub-Saharan Africa [2, 15]. However, the contribution of malaria to anaemia in our study population was small as only 6 patients (3.5%) had malaria parasites on enrolment. This is in line with data from other studies that show that the role of malaria in causing anaemia, especially in HIV patients in sub-Saharan Africa [2, 14], is limited and likely to have been overestimated [31, 33].

Malnutrition was common in our population as half the patients had a BMI below 18.5m$^2$ and deficiencies of iron and folate were diagnosed in 33.9% and 12% of the patients

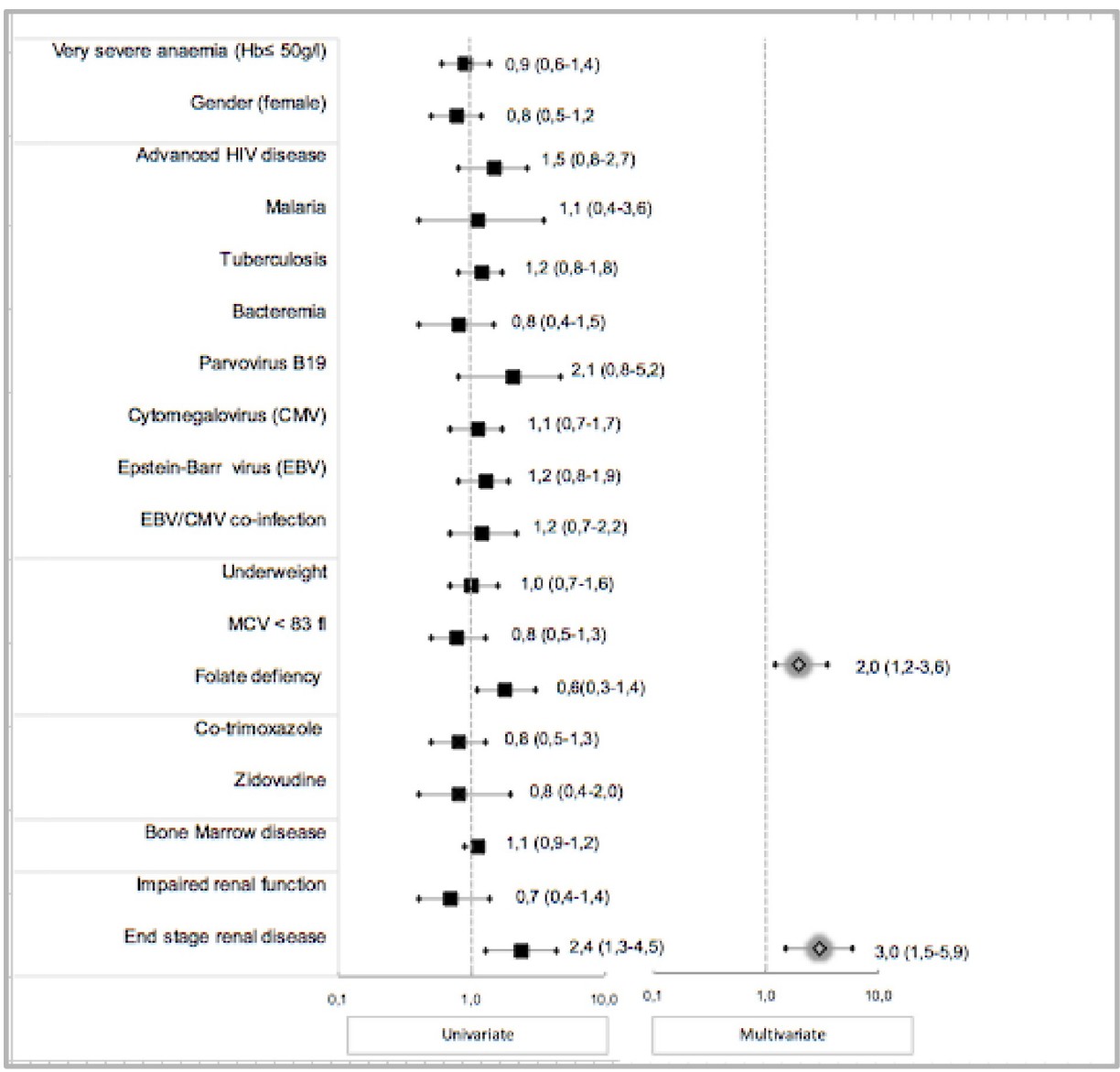

**Fig 3. Risk factors for 365-day mortality in HIV-infected patients with severe anaemia.** Univariate and multivariate Cox regression outcome (Hazard Ratios 95% CI). Folate deficiency (≤3 ng/l) HR 2.0 95% CI 1.2–3.6 and end stage renal disease (GFR ≤15); HR 3.0 5% CI 1.5–5.9, were associated with overall mortality. Abbreviations: Hb: Haemoglobin, GFR; Glomerular filtration rate, VL Viral Load.

respectively. The prevalence of iron deficiency (MCV ≤ 83 fl) in our study is higher than the 18%-25% prevalence reported in anaemic HIV-infected patients in sub-Saharan Africa reported in previous studies [50, 51]. Interestingly, children and adults in sub-Saharan Africa with severe anaemia but without HIV infection, have a higher prevalence of iron deficiency of respectively 59% and 47%, than we found in our HIV-infected population [30,33]. Previous studies on severe anaemic HIV patients were published before 2010; at that time ART availability was limited and all the studies included HIV-infected patients who were ART naïve and who had high rates of immune suppression. HIV-infected patients who are in the early stages of infection or who are on effective treatment may have better immune function so in these patients the aetiology of severe anaemia (e.g. iron deficiency) may be more similar to that of

non-HIV infected patients [33]. Iron deficiency diagnosis is complicated by a lack of peripheral iron markers and microcytosis (i.e. low MCV) poorly reflects bone-marrow iron deficiency especially in the context of HIV-infection (Huibers et al PLOS One, 2019). Bone marrow evaluation remains the golden standard for diagnosis of iron deficiency. In addition to finding a high prevalence of iron deficiency (MCV g 83 fl), ours is the first study to document a possible association between folate deficiency and increased mortality in severely anaemic HIV-infected patients. There are a limited number of studies describing micronutrient supplementation, including folate, for anaemic HIV-infected adults, but overall supplementation appears to have little effect on reducing morbidity and mortality [43,52]. Macronutrient support using, for example, fortified wheat flour, has had beneficial effects on anaemia reduction and micronutrient levels in populations in sub-Saharan Africa, but has never been tested in the context of severe anaemia in HIV-infected patients [53,54]. More research is therefore needed to evaluate the effectiveness of both macro- and micro- nutritional support in severely anaemic HIV-infected adults.

HIV-infected adults have an increased risk of neoplastic bone marrow diseases, which can often cause anaemia [14]. However, prevalence data for such conditions among HIV-infected patients with anaemia, especially in resource limiting settings, are scarce. Only three of our study patients had confirmed bone marrow malignancies. In contrast, and in line with previous studies, MDS was common occurring in 27% of our patients [28,55].

Renal impairment was a frequent finding among our study patients and has been linked to HIV disease progression, anaemia and poor outcomes in both wealthy and resource limited settings [56–59]. Evaluation of renal function is an important component of severe anaemia treatment protocols [34, 57] since it may affect the choice of ART [56, 60]. It may also have implications for clinical management, for example by introducing measures to prevent further deterioration or considering the use of erythropoietin[57].

Our study has several limitations. We purposely did not explore factors potentially associated with anaemia that previous studies had shown were uncommon in Malawi, such as haemoglobinopathies and parasitic infections [33, 61]. We also did not include an HIV-infected population without severe anaemia against which to compare the prevalence of anaemia aetiologies and clinical outcomes. Only a sub-population of our patients gave consent for bone marrow sampling. Although these were an unselected group of patients it is possible that this may have introduced a bias, for example by excluding the patients who were particularly unwell. Nevertheless, our findings are very valuable since bone marrow data from HIV-infected African patients is very scarce.

## Conclusion

Our study has demonstrated that severe anaemia in HIV-infected adults in Malawi is associated with multiple co-existing aetiologies and has a strikingly high mortality rate. Severe anaemia in HIV-infected patients is therefore a critical indicator of mortality and requires urgent and multiple interventions. Particularly important are the initiation of ART, the management of infections such as TB and CMV, and optimisation of renal function. Intervention studies are needed to properly define the role and safety of iron and folate supplementation, as well as to develop and evaluate guidelines, which are feasible in resource-limited settings to help clinicians manage these patients more effectively.

## Supporting information

**S1 Data.**
(DTA)

**S1 Fig.**
(TIF)

**S1 Table.**
(DOCX)

**S2 Table.**
(DOCX)

## Acknowledgments

The authors would like to thank all of the study participants, doctors, nurses and support staff of Queens Elizabeth Hospital and the Malawi-Liverpool-Wellcome centre in Blantyre for their participation and cooperation. This study was supported by the Nutricia Research Foundation (Project number 2017–43), The Hague, the Netherlands and the Wellcome Trust (Project number WT086559), Liverpool, United Kingdom. The funders had no role in the study design, data collection and analysis, decision to publish or preparation of the manuscript.

## Author Contributions

**Conceptualization:** Minke H. W. Huibers, Imelda Bates, Steve McKew, Job C. Calis.

**Data curation:** Minke H. W. Huibers, Steve McKew, Theresa J. Allain, Chimota Phiri.

**Formal analysis:** Minke H. W. Huibers, Imelda Bates, Sarah E. Coupland.

**Funding acquisition:** Minke H. W. Huibers, Steve McKew.

**Methodology:** Imelda Bates, Kamija S. Phiri, Job C. Calis.

**Project administration:** Chimota Phiri.

**Supervision:** Imelda Bates, Michael Boele van Hensbroek, Job C. Calis.

**Writing – original draft:** Minke H. W. Huibers.

**Writing – review & editing:** Imelda Bates, Theresa J. Allain, Sarah E. Coupland, Kamija S. Phiri, Michael Boele van Hensbroek, Job C. Calis.

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
