## [Decision Letter · Decision Letter 0]

17 Jul 2019

PONE-D-19-15827

Severe anaemia complicating HIV in Malawi; multiple co-existing aetiologies are associated with high mortality.

PLOS ONE

Dear mrs Huibers,

Thank you for submitting your manuscript to PLOS ONE. After careful consideration, we feel that it has merit but does not fully meet PLOS ONE’s publication criteria as it currently stands. Therefore, we invite you to submit a revised version of the manuscript that addresses the points raised during the review process.

Among them, particular attention should be given on comparing patients who died early, later or survived, and also on comparing patients on ART or not.

We would appreciate receiving your revised manuscript by Aug 31 2019 11:59PM. To enhance the reproducibility of your results, we recommend that if applicable you deposit your laboratory protocols in protocols.io, where a protocol can be assigned its own identifier (DOI) such that it can be cited independently in the future. For instructions see: http://journals.plos.org/plosone/s/submission-guidelines#loc-laboratory-protocols

We look forward to receiving your revised manuscript.

Kind regards,

Kostas Pantopoulos, PhD

Academic Editor

PLOS ONE

Journal Requirements:

1. Please note that all PLOS journals ask authors to adhere to our policies for sharing of data and materials: https://journals.plos.org/plosone/s/data-availability. According to PLOS ONE’s Data Availability policy, we require that the minimal dataset underlying results reported in the submission must be made immediately and freely available at the time of publication. As such, please remove any instances of 'unpublished data' or 'data not shown' in your manuscript and replace these with either the relevant data (in the form of additional figures, tables or descriptive text, as appropriate), a citation to where the data can be found, or remove altogether any statements supported by data not presented in the manuscript.

Reviewers' comments:

Reviewer's Responses to Questions

**Comments to the Author**

1. Is the manuscript technically sound, and do the data support the conclusions?

Reviewer #1: Partly

Reviewer #2: Yes

2. Has the statistical analysis been performed appropriately and rigorously? 

Reviewer #1: Yes

Reviewer #2: Yes

3. Have the authors made all data underlying the findings in their manuscript fully available?

Reviewer #1: No

Reviewer #2: Yes

4. Is the manuscript presented in an intelligible fashion and written in standard English?

Reviewer #1: Yes

Reviewer #2: Yes

5. Review Comments to the Author

Reviewer #1: General comments:

It is well established that anaemia and anaemia severity are associated with HIV disease stage and mortality. HIV-associated anaemia has many potential causes. In this manuscript, Huibers et al report the prevalence of diverse co-existing morbidities that can be linked to anaemia in a cohort of hospitalised HIV patients presenting with severe anaemia (Hb<7g/dL, N=199 – median 5.3g/dL), many of whom were immunosuppressed. They classified participants as having severe anaemia or very severe anaemia (Hb<5g/dL) and tested, using univariate and multivariate analyses, whether there were differences in risk of co-morbidities according to anaemia severity; they found some evidence of increased risk of CMV-EBV infection in the very severe anaemia group. The cohort had a high mortality rate, presumably reflecting a high prevalence of advanced HIV disease at hospital presentation. Mortality was associated with end-stage renal disease and (more weakly) with folate deficiency; anaemia severity was not associated with mortality risk (bearing in mind all participants had Hb<7g/dL according to enrolment criteria).

Similar studies from the same location have previously been published (e.g. Lewis et al, Trans R Soc Trop Med Hyg. 2005) considering multiple-potential aetiologies of anaemia in hospitalised patients. The data in the present study are largely descriptive, focussing purely on HIV-infected patients; they are robust, although the inclusion criterion of Hb<7g/dL does limit evaluation to comparison of severe vs very severe anaemia (as opposed to e.g. no anaemia/mild anaemia). The discussion is long and could be made more concise. In places it seems to infer a causative relationship between anaemia and mortality, but the present analyses do not allow such inferences to be made; HIV pathogenesis is complex, and although anaemia may contribute to morbidity and mortality, the data show that there are many other co-morbidities that could also explain the mortality risk.

This manuscript was submitted with a parallel manuscript evaluating markers of iron status in a subset of the above cohort of severely anaemic HIV-infected patients, to be evaluated concurrently.

Specific points:

1. Data availability: the authors state that raw data will be available within the manuscript or supplementary information – it’s not clear whether this has been provided yet. It would be helpful to refer to this within the manuscript.

2. Introduction (paragraph 2): it would be useful to include brief consideration of how important anaemia is clinically in relation to other co-morbidities (i.e. how important is correction of anaemia per se).

3. Introduction: The authors should specifically mention inflammation as a potential cause of anaemia (this will likely be present during many of the infections, but may not be restricted to these).

4. Methods: the authors confirm whether blinding was used in bone marrow analyses.

5. Methods/Results/Discussion (line 83): iron deficiency – the parallel manuscript demonstrates that MCV is a poor index of iron status (bone marrow iron staining, considered gold standard). The authors should make this clear in the present study in the text / table footnotes (e.g. by referring to the second manuscript) – it is therefore unclear how useful this part of the analysis is. Could the authors incorporate bone marrow iron status into analysis, for the subset of patients for whom that information is available?

6. Results – Table 2:

o % with severe anaemia (in title row) should be corrected.

o In the table footer, the authors should clarify that analyses are comparing severe with very severe anaemia.

o The methods section states that variables with P<0.1 in univariate are included in multivariate analysis: it appears that variables with P>0.1 are included – this should be clarified / corrected.

o Renal function: normal – is it intentional that no statistics are included here?

7. Results:

o do particular co-morbidities frequently occur together?

o do any particular combinations associate with anaemia severity, haemoglobin concentration or mortality/mortality within first 60 days?

8. Results: Table 1 / Text: there is a discrepancy between the mortality reported in the text and in the Table. This should be corrected.

9. Results: Given the pattern of mortality (approx. 50% within 1 year, approx. 80% of whom died within 60 days):

o The analysis could be refined to compare patients who died within 60 days vs those who died later and/or those who survived: was there a difference in anaemia group, haemoglobin concentration, or co-morbidities between these groups?

10. Results: Figure 3 – the position of the “zero” line should be aligned with 0.

11. Discussion:

o In general the discussion can be made more concise, e.g. see repetition in lines 10-11 and 20-21.

o Lines 14-15: this could be interpreted as anaemia being causal for mortality; it is part of a complex disease state. Figure 3 found no association of anaemia severity with mortality risk (noting that all individuals had at least severe anaemia). However, in concert with much previous literature, the data are consistent with a associations between severe anaemia and mortality in HIV infection given this mortality rate. Suggest revising to “a multi-causal critical condition associated with high mortality”.

o Line 26-27: it is a little unclear whether the authors are implying that severe anaemia is itself a prime cause of mortality (that therefore should be addressed per se), or rather whether it is a clear marker of severe disease (line 30) that would likely be improved through addressing co-existing conditions associated with advanced HIV (infections etc). The literature cited regarding anaemia and mortality is correlative, so the latter interpretation is favoured in my view.

o Line 34: add “and inflammation”?

o Line 36: the lack of effect here could reflect all patients having Hb<7g/dL in the present study.

o Line 39-41: the authors could cite relevant data on the impact of ART on anaemia during HIV infection.

o Line 49 – refer to point on line 26-27.

o Line 83-87: see comment above regarding iron deficiency definition and reference to the second manuscript; the prevalence of BM-ID in the subset of individuals assessed in the second manuscript was higher (48%) and could be cross-referenced here. Should Line 87 use ref 46, not 45? Reference 46 used BM iron rather than MCV as the definition.

o Line 92-94: Caution should be employed in making this inference – these patients are still all severely anaemic with multiple co-morbidities, and a high proportion are still immunosuppressed. It is possible the discrepancies are more likely related to assessment of iron status.

Formatting / Spelling:

• Abstract: repetition of “of anaemia” in Methods paragraph.

• Discussion: line 39: “therefore”

• Discussion: line 91: “severe”

• Discussion: line 92: “anaemia” not “anaemic"

• Discussion: line 94: “conditions such as iron deficiency are….”

Reviewer #2: i would like to thank the authors for submitting this article for submission. The article highlights the very important problem of severe anemia in HIV infected adults in Malawi and the multiple etiologies associated with this diagnosis. i believe this article contributes something new to the body of evidence already available for decision makers. I would like to recommend this article for publication but first I would like to authors to address a few concerns.

Major comments

in the methods section on page 10, the authors note that all HIV infected patients over 18 years old admitted to the general medical ward with severe anemia were approached and recruited for the study if they provided informed consent. How was the sample size (n=199) determined? this was not explained in the methods section.

How did the authors come about the 15 initial etiological factors that were explored for possible associations with severe anemia in HIV infected adults? some of the factors listed such as malaria, zidovudine use, iron deficiency have long been associated with anemia but others such as EBV, CMV, vitamin B12 not so much. Could the authors expand on this further in the introduction, discussion?

More than half of the study population was not on ART with low CD4 counts and high viral loads- thus this study population may not be generalizable to other adult HIV-infected populations in resource poor countries as the majority of the population were not on standard ART treatment and had a higher risk of mortality. the study would be more informative if the outcomes were looked at separately in patients on ART and those not on ART to determine if there was a significant difference in mortality, as well as in the associations with the different etiological factors.

Minor comments:

Figure 1: the title and explanation of the figure is in the manuscript but not on the figure itself. Similar comment for Figure 2, readers should be able to understand the figures by looking at them without having to refer to the manuscript for detailed explanation.

in figure 3, what do the plotted numbers represent and how will the readers interpret them? are they median values with interquartile ranges or hazard ratios with 95% confidence intervals? this was explained in the manuscript but should also be clear from looking at the figure itself.

6. PLOS authors have the option to publish the peer review history of their article (what does this mean?). If published, this will include your full peer review and any attached files.

Reviewer #1: No

Reviewer #2: No

---

## [Author Response · Author response to Decision Letter 0]

2 Oct 2019

To: PLOS ONE Kostas Pantopoulos, PhD Academic Editor

Subject: Rebuttal letter resubmission Manuscript ID: PONE-D-19-15827

September 2019

Dear Kostas Pantopoulos PLOS ONE,

We thank you for considering our manuscript titled; “Severe anaemia complicating HIV in Malawi; multiple co-existing aetiologies are associated with high mortality” (PONE-D-19-15827) for publication in your journal. We would like to highlight that we initially submitted two manuscripts back to back. The other manuscript titled; “Hepcidin and conventional markers to detect iron deficiency in severely anaemic HIV-infected patients in Malawi” (PONE-D-19-15824) has also been resubmitted. Please receive the suggested changes and answers considering the issues raised by the reviewer. In our reply we indicated the original text (with respective line and page numbers) and changes are underlined. As requested we have tracked change all changes made in the original manuscript.

Major comments:

General. Data availability:

1. All data was presented in the manuscript wit the exception the notification in line 393 (page 15): “TB medication itself was not associated with the severity of anaemia (data not shown).” 

• We adjusted this and provided the data directly. TB medication itself was not associated with the severity of anaemia (OR 1.6 95%CI 0.7-3.8).

• All data was given within the manuscript as described above. We did not explicitly ask for medical ethical permission to publish our database online. In case individual readers are interested in using anonymised data we can and will share our database or requested data, we have added this to the results section 

• Methods were adjusted line 214 (page 7; Additional raw data can be requested by contacting the corresponding author. 

Ad reviewer 1:

Introduction

2. It would be useful to include brief consideration of how important anaemia is clinically in relation to other co-morbidities (i.e. how important is correction of anaemia per se).

• We have added the requested item to the introduction. Line 63(Page3). Anaemia treatment may even improve survival (1). 

3. The authors should specifically mention inflammation as a potential cause of anaemia (this will likely be present during many of the infections, but may not be restricted to these).

• We agree with the reviewer. Line 66-68 (page3) introduction were rephrased: Severe anaemia in HIV infection has been associated with micronutrient deficiencies, infections (viral, bacterial and parasitic) and inflammation, medication induced (Zidovudine and co-trimoxazole), neoplastic diseases (7-11).

Methods

4. The authors confirm whether blinding was used in bone marrow analyses.

• We confirm that blinding was used. The method section has been adjusted: line 159 (page 5): Bone marrow analysis was performed by examiners that were blinded to patient data.

5. Methods/Results/Discussion (line 83): iron deficiency – the parallel manuscript demonstrates that MCV is a poor index of iron status (bone marrow iron staining, considered gold standard). The authors should make this clear in the present study in the text / table footnotes (e.g. by referring to the second manuscript) – it is therefore unclear how useful this part of the analysis is. Could the authors incorporate bone marrow iron status into analysis, for the subset of patients for whom that information is available? 

• We specifically chose the MCV as it a) was the best (suboptimal) conventional marker in our setting b) was available in more patients than the subgroup of patients with a bone marrow and c) is available in most African settings and therefor applicable to the settings most interested readers work in. We have added the requested data however to the footnote of table 1. 

• Methode line 178 (page 6): In a sub study (Huibers et al PLOSone2019) we evaluated bone marrow iron deficiency toward several conventional blood makers; MCV (fl), MCH (pg/cells), Fe (umol/l), Ferritine (ug/dl), TFr1 receptor (nmol/l), TrF index (stFR/Log ferritine). All markers showed suboptimal correlations AUCROC < 0.6 with bone-marrow iron deficiency. Of the conventional markers MCV performed best; AUCROC of 0.545. MCV was used to identified iron deficiency as it was the best (suboptimal) conventional marker in our setting, was available in more patients than the just the subgroup (n=76) of patients with a bone marrow result plus is available and known in most African settings (3, 22, 30).

• Footnote Table 2. Distribution and multivariate analysis of co-existing factors associated with severe (Hb≤ 70 g/l- (Hb> 50 g/l) versus very severe anaemia (Hb≤ 50 g/l) in HIV-infected adults in Malawi. 1 A total of 28-blood cultures were positive, the most common organisms were E. coli (42.9%; 12/28) and non-Typhoid Salmonella (17.9 %; 5/28). 2 Iron deficiency was defined by MCV≤ 83 fl; In a sub study (Huibers et al PLOSone2019) we evaluated bone marrow iron deficiency toward several conventional blood makers; MCV (fl), MCH (pg/cells), Fe (umol/l), Ferritine (ug/dl), TFr1 receptor (nmol/l), TrF index (stFR/Log ferritine). All markers showed suboptimal correlations AUCROC < 0.6 with bone-marrow iron deficiency. Of the conventional markers MCV performed best; AUCROC of 0.545. 

• Line 449 (page 16) discussion was rephrased: Iron deficiency is complicated by a lack of peripheral iron markers and diagnosing remains challenging as peripheral blood markers as MCV poorly reflect bone-marrow iron deficiency especially in the context of HIV-infection (Huibers et al PLOSone 2019). 

Results 

6. Table 2: - % with severe anaemia (in title row) should be corrected. 

• This has been adjusted (line 388 page 11): Overall N=199(100%) and severe anaemia N=115/199 (57.8%) 

7. Table 2: In the table footer, the authors should clarify that analyses are comparing severe with very severe anaemia. 

• Rephrased (line 398 page 12): Table 2. Distribution and multivariate analysis of co-existing factors associated with severe (Hb≤ 70 g/l- Hb> 50 g/l) versus very severe anaemia (Hb≤ 50 g/l) in HIV-infected adults in Malawi. 

8. The methods section states that variables with P<0.1 in univariate are included in multivariate analysis: it appears that variables with P>0.1 are included – this should be clarified / corrected.

• We have accidentally added a VL > 1000 copies with a p-value of 0.163 in the multivariate analysis as well. Wherefore excuse. We adjusted table 2 accordingly. Multivariate outcome did not change significantly (line 398 page 12). 

9. Renal function: normal – is it intentional that no statistics are included here?

• In the Logistic regression impaired function and end stage renal function were compared toward a normal renal function. We agree with the reviewer that we displayed this in a confusing way. To display the results more clearly we deleted the normal renal function from table two (line 398 page 12).

10. Do particular co-morbidities frequently occur together?

• There is obvious overlap of these variables as is graphically displayed in the new supplementary figure S1. Given that this overlap and the wide spread, we have chosen not to over interpretate our dataset and have restricted our analysis to simple uni- and multivariate analyses. 

• The results section was adjusted line 247 (page9) Overlap of risk factors is visually displayed in supplemental figure S1.

11. Given the pattern of mortality (approx. 50% within 1 year, approx. 80% of whom died within 60 days). The analysis could be refined to compare patients who died within 60 days vs those who died later and/or those who survived: was there a difference in anaemia group, haemoglobin concentration, or co-morbidities between these groups?

• We appreciate this question from the reviewer. To display this information we created a supplementary table S1. The prevalence of EBV was significant different between the group of patients who died within 60 days (50%) and after 60 days (12.5%); p-value 0.006. However when we do a multivariate cox regression for mortality EBV is not significant related toward mortality either within 60 as after 60 days. We therefore did not wanted to include the outcome of this table directly in the results section. But with regard to the question we added the supplemental table S1. Results of the uni and multivariate analysis are aswell provided in supplementary table S2. See comment below.

• Methods were adjusted in line 204 (page 6). Moreover a sub analysis was performed to evaluate the distribution of co-existing factors between mortality within and after 60 days. Group comparisons for categorical data were performed using the χ2 test or Fisher’s exact test, and for continuous data using the t-test or the Wilcoxon rank-sum test.

• Results were adjusted in line 302 (page 10): Distribution of co-existing factors between mortality within 60 days and overall mortality in HIV-infected adults in Malawi was given in supplementary table S1

• See for rephrasing on table S2 point 12. 

12. Do any particular combinations associate with anaemia severity, haemoglobin concentration or mortality/mortality within first 60 days?

• We included outcome in the result section in an additional supplemental figure and results were rephrased (Line 306 page 10): Folate deficiency and end stage renal disease were associated with overall mortality with Hazard Ratio 2.2 (95% CI 1.2-3.8) and Hazard Ratio 3.2 (95% CI 1.6-6.2) respectively (figure 3). End stage renal disease (Hazard Ratio 2.7 95% CI 1.2-6.2) was associated with mortality within 60 days, supplementary table S2. 

• Secondly this sub-analysis was added to the method Methods were adjusted in line 207 page: Uni- and multivariate analyses were done using logistic regression and Cox regression to describe predictors of overall mortality. A sub- analysis was performed for mortality within 60 days. 

13. Table 1 / Text: there is a discrepancy between the mortality reported in the text and in the Table. This should be corrected.

• This was corrected. Line 300 page 10: During the one-year follow-up period, 101 study patients (50.8%) died. The median time to death was 17.5 days (IQR 6-55) and 81 (80.2%) of these deaths occurred within 60 days of admission (figure 2). 

14. Results: Figure 3 – the position of the “zero” line should be aligned with 0.

• We have adjusted the lines to 1.0 (the neutral OR)

• With adjusting the neutral OR we found a minor error, which we corrected. The text described a minimally different outcome, difference in decimal (HR/CI) than shown in figure 3. This has been adjusted. Overall outcome did not change. 

• Folate deficiency (≤3 ng / l); HR 2.1 95% CI 1.2-3.6 → HR 2.0 (95% CI 1.2-3.6) and stage renal disease (GFR ≤15); HR 3.2 5% CI 1.6-6.2) → HR 3.0 5% CI 1.5-5.9. 

• Line 303 page 10: Folate deficiency and end stage renal disease were associated with overall mortality with Hazard Ratio 2.0 (95% CI 1.2-3.6) and Hazard Ratio 3.0 (95% CI 1.5-5.9) respectively (figure 3).

• Figure 3. Risk factors for 365-day mortality in HIV-infected patients with severe anaemia. Univariate and multivariate Cox regression outcome (Hazard Ratios 95% CI). Folate defiency (≤3 ng/l) HR 2.0 95% CI 1.2-3.6 and end stage renal disease (GFR ≤15); HR 3.0 5% CI 1.5-5.9, were associated with overall mortality.

Discussion

The discussion is long and could be made more concise. In places it seems to infer a causative relationship between anaemia and mortality, but the present analyses do not allow such inferences to be made; HIV pathogenesis is complex, and although anaemia may contribute to morbidity and mortality, the data show that there are many other co-morbidities that could also explain the mortality risk.

15. We have rephrased several section of the discussion to comply with these comments:

• Discussion line 455, page 14: Severe anaemia in HIV-infected patients in a resource limited setting, such as Malawi, is a multi-causal critical condition associated with high mortality.

• Discussion line 459, page 14 rephrased: However HIV pathogenesis is complex and although anaemia may contribute to morbidity and mortality, the data suggest that the potential aetiology is multi-causal. Previous studies reported an estimated one-year mortality of to 30-55% in severe anaemic versus 3.7% in non-anaemic HIV-infected patients in resource limited settings (5, 6)

• Conclusion underlines the opinion of the reviewer: Line 642, page 17: Severe anaemia in HIV-infected patients is therefore a critical indicator of mortality and requires urgent and multiple interventions. “ That we emphasise that criticality of anaemia without phrasing the direct and only contributor of mortality in the complex HIV-pathophysiology. 

16. In general the discussion can be made more concise, e.g. see repetition in lines 10-11 and 20-21. We have made several adjustments. All adjustments are explained and displayed in the comments below. 

• Discussion line 457, page 14: Anaemia in HIV-infected patients is descripted as an independent predictor of mortality with a direct effect of decreasing haemoglobin concentrations. 

• Discussion line 463, page 14: The seriousness of severe anaemia and its complexity in HIV-infected patients needs to be better recognized.

17. Lines 14-15: this could be interpreted as anaemia being causal for mortality; it is part of a complex disease state. Figure 3 found no association of anaemia severity with mortality risk (noting that all individuals had at least severe anaemia). However, in concert with much previous literature, the data are consistent with an associations between severe anaemia and mortality in HIV infection given this mortality rate. Suggest revising to “a multi-causal critical condition associated with high mortality”.

• See comment 15, rephrasing Discussion line 455, page 14.

18. Line 26-27: it is a little unclear whether the authors are implying that severe anaemia is itself a prime cause of mortality (that therefore should be addressed per se), or rather whether it is a clear marker of severe disease (line 30) that would likely be improved through addressing co-existing conditions associated with advanced HIV (infections etc). The literature cited regarding anaemia and mortality is correlative, so the latter interpretation is favoured in my view.

• We appreciated this comments and adjusted the discussion, line 334, page14 is rephrased: The seriousness of severe anaemia and its complexity in HIV-infected patients needs to be better recognized. Irrespective if severe anaemia is the direct cause of the mortality or merely linked to other contributors causing both death and anaemia, one can state that severe anaemia should trigger clinicians as an emergency sign and therefor should be grouped as a stage 4 condition in stead of the current stage 3 classification (33). 

19. Line 34: add “and inflammation”?

• We agree with the reviewer and adjusted discussion line 471, page 14: HIV may cause anaemia directly through an inhibitory effect of the HIV-virus on the erythropoietin progenitor cells in the bone marrow, or indirectly through opportunistic infections and/or inflammation causing anaemia (37).

20. Line 36: the lack of effect here could reflect all patients having Hb<7g/dL in the present study.

• We agree with the reviewer and adjusted discussion line 474, page 14: In our cohort this was not confirmed however our cohort consist with patients who all experienced severe anaemia. 

21. Line 39-41: the authors could cite relevant data on the impact of ART on anaemia during HIV infection.

• Discussion Line 555, page 15 was adjusted: Therefore they are likely to continue having high levels of life-threatening anaemia, as the complex multifactorial ethology of the severe anaemia might be marker of sever HIV disease. At last the effect of medication as Zidovidine and trimoxazole on the bone marrow should also be taken in consideration. However possible due to the bias of all severe anaemic HIV-infected patients, this was not confirmed as directly effective on severity of anaemia.

22. Line 49 – refer to point on line 26-27.

• We appreciate this suggestion. Discussion line 553, page 15 is rephrased; Consequently they are likely to continue having high levels of life-threatening anaemia, as the complex multifactorial ethology of the severe anaemia might be marker of sever HIV disease. 

23. Line 83-87: see comment above regarding iron deficiency definition and reference to the second manuscript; the prevalence of BM-ID in the subset of individuals assessed in the second manuscript was higher (48%) and could be cross-referenced here. Should Line 87 use ref 46, not 45? Reference 46 used BM iron rather than MCV as the definition.

• See point 5 Line 449 (page 16) discussion was rephrased: Iron deficiency is complicated by a lack of peripheral iron markers and diagnosing remains challenging as peripheral blood markers as MCV poorly reflect bone-marrow iron deficiency especially in the context of HIV-infection (Huibers et al PLOSone 2019). 

• We agree with the reviewer. Reference 46 was replaced by 45

24. Line 92-94: Caution should be employed in making this inference – these patients are still all severely anaemic with multiple co-morbidities, and a high proportion are still immunosuppressed. It is possible the discrepancies are more likely related to assessment of iron status

• Discussion line 615-616, page 16 was rephrased: In this group of HIV-infected patients with better immune systems the aetiology of severe anaemia may be more similar to the multifactorial aetiology of non-HIV infected patients. Conditions such as iron deficiency are more alike the HIV-uninfected groups.

Formatting / Spelling:

25. Abstract: repetition of “of anaemia” in Methods paragraph. 

• Rephrased: Fifteen potential causes and associations with anaemia severity and mortality were explored.

26. Discussion: line 39: “therefore” 

• This line was already rephrased with regard to above comments. Discussion line 525 page 15: Consequently they are likely to continue having high levels of life-threatening anaemia, as the complex multifactorial ethology of the severe anaemia might be marker of sever HIV disease.

27. Discussion: line 91: “severe” | line 92: “anaemia” not “anaemic" line 94: “conditions such as iron deficiency are….” 

• Rephrased discussion line 587-588, page 16 “In this group of HIV-infected patients with better immune systems the aetiology of severe anaemia may be more similar to the aetiology of non-HIV infected patients. Conditions such as iron deficiency are more alike the HIV-uninfected groups.

Reviewer 2;

Major comments

28. Sample size: in the methods section on page 10, the authors note that all HIV infected patients over 18 years old admitted to the general medical ward with severe anemia were approached and recruited for the study if they provided informed consent. How was the sample size (n=199) determined? this was not explained in the methods section.

• Methods- statistics were adjusted. Methode line 193 (page 6). The study was primarily designed to give a complete overview of the potential factors associated with severe anaemia with a special attention of the role of lymphomas in a severe anemic african patients ipopulation infected with HIV. A sample size of 200 would be able to detect an estimated prevalence of 5 or 10% with confidence intervals of 2.4% - 9% and 6- 15% respectively and was therefor chosen. 

29. How did the authors come about the 15 initial etiological factors that were explored for possible associations with severe anemia in HIV infected adults? some of the factors listed such as malaria, zidovudine use, iron deficiency have long been associated with anemia but others such as EBV, CMV, vitamin B12 not so much. Could the authors expand on this further in the introduction, discussion?

• The 15 potential factors studied were based on two previous studies performed in Malawi and a systemic review on this topic:

• Calis JC, Phiri KS, Faragher EB, Brabin BJ, Bates I, Cuevas LE, et al. Severe anemia in Malawian children. N Engl J Med. 2008;358(9):888-99.

• Lewis DK, Whitty CJ, Walsh AL, Epino H, Broek NR, Letsky EA, et al. Treatable factors associated with severe anaemia in adults admitted to medical wards in Blantyre, Malawi, an area of high HIV seroprevalence. Transactions of the Royal Society of Tropical Medicine and Hygiene. 2005;99(8):561-7.

• Calis JC, van Hensbroek MB, de Haan RJ, Moons P, Brabin BJ, Bates I. HIV-associated anemia in children: a systematic review from a global perspective. AIDS. 2008 Jun 19;22(10):1099-112. 

• This comment was rephrased in the method section line 168 (page 5): A total of 15 potential factors involved in the aetiology of severe anaemia were investigated, factors were based on two previous studies performed in Malawi and a systemic review on this topic (30-32). 

30. More than half of the study population was not on ART with low CD4 counts and high viral loads- thus this study population may not be generalizable to other adult HIV-infected populations in resource poor countries as the majority of the population were not on standard ART treatment and had a higher risk of mortality. The study would be more informative if the outcomes were looked at separately in patients on ART and those not on ART to determine if there was a significant difference in mortality, as well as in the associations with the different etiological factors.

• We agree that it appears informative to stratify our analyses for ART use at enrolment, which applied to nearly half the study population. In both the mortality (overall and 60 days) and the severity analyses ART use at enrolment was not associated with these outcomes. Therefore we decided not to present a stratified analysis in this paper. In line 403 of the discussion we further highlight the fact that the findings of our study remain very relevant because many HIV-infected patients in resource-limited settings present late in the course of their disease or are unable to access reliable supplies of ART. This may especially apply to those with severe anaemia, as it is a complication of advanced disease.

Minor comments:

31. Figure 1: the title and explanation of the figure is in the manuscript but not on the figure itself. 

• Figure 1. Total number aetiologies for severe anaemia co-existing in each patient (n=199). Mean is 3 factors (SD 1.3), range 1-8. Aetiologies for severe anaemia include: 1) Unsuppressed HIV-infection; viral load ≥1000 copies/ml. 2) TB: one or more of the following were present: a) positive sputum culture, b) chest X-ray with signs of pulmonary TB and/or c) on going TB treatment at time of enrolment d) clinical diagnosis by local doctor including unknown generalized lymphadenopathy and/or night sweats of > 30 days and of unknown origin e) caseating granulomata in the bone marrow trephine. 3) Malaria: presence of malaria parasites in a thick blood film. 4) Parvovirus B19: viral load of >1000 copies/ml. 5) Cytomegalovirus (CMV); load of >100 copies/ml. 6) Epstein-Barr virus (EBV); viral load >100 copies/ml. 7) Bacteraemia; a blood culture growing a potential pathogen. 8) Underweight (BMI ≤18.5). 9) Serum folate deficiency (≤3 ng/l). 10) Vitamin B12 deficiency (≤180 pg/ml). 11). Iron deficiency was defined as MCV≤ 83 fl (3, 22, 33). 12) Zidovudine usage. 13) Co-trimoxazole usage. 14) Bone marrow disorders; lympho-proliferative disease, myeloid-proliferative disease or MDS. 15) Renal impairment: a GFR which either indicated impaired (GFR 15–59 ml/min/1.73 m2) or End Stage (GFR ≤15 ml/min/1.73 m2) Renal Disease (21, 34).

32. Similar comment for Figure 2, readers should be able to understand the figures by looking at them without having to refer to the manuscript for detailed explanation.

• Line 385 page 10; Figure 2. Kaplan Meyer survival curve over time (days) for adult Malawian patients with HIV infection and severe anaemia during 365 days follow-up. Abbreviation: 95% confidence interval (95%CI).

33. Figure 3, what do the plotted numbers represent and how will the readers interpret them? are they median values with interquartile ranges or hazard ratios with 95% confidence intervals? 

• Line 388 page 10; Rephrased: Figure 3. Risk factors for 365-day mortality in HIV-infected patients with severe anaemia. Univariate and multivariate Cox regression outcome (Hazard Ratios 95% CI). 

We hope to have clarified outstanding questions and improved the manuscript according to the concerns raised. Please do not hesitate to contact us if you have any further questions. 

With many thanks for your consideration, and on behalf of all the authors.

Minke Huibers, MD

---

## [Decision Letter · Decision Letter 1]

23 Oct 2019

PONE-D-19-15827R1

Severe anaemia complicating HIV in Malawi; multiple co-existing aetiologies are associated with high mortality.

PLOS ONE

Dear mrs Huibers,

Thank you for submitting your manuscript to PLOS ONE. Both reviewers found that the revised manuscript is improved and has addressed most of their concerns. However, reviewer 1 raised a few remaining minor issues, which require attention. Therefore, we invite you to address them in a revised version of the manuscript.

We would appreciate receiving your revised manuscript by Dec 07 2019 11:59PM. To enhance the reproducibility of your results, we recommend that if applicable you deposit your laboratory protocols in protocols.io, where a protocol can be assigned its own identifier (DOI) such that it can be cited independently in the future. For instructions see: http://journals.plos.org/plosone/s/submission-guidelines#loc-laboratory-protocols

We look forward to receiving your revised manuscript.

Kind regards,

Kostas Pantopoulos, PhD

Academic Editor

PLOS ONE

Reviewers' comments:

Reviewer's Responses to Questions

**Comments to the Author**

1. If the authors have adequately addressed your comments raised in a previous round of review and you feel that this manuscript is now acceptable for publication, you may indicate that here to bypass the “Comments to the Author” section, enter your conflict of interest statement in the “Confidential to Editor” section, and submit your "Accept" recommendation.

Reviewer #1: (No Response)

Reviewer #2: All comments have been addressed

2. Is the manuscript technically sound, and do the data support the conclusions?

Reviewer #1: Partly

Reviewer #2: Yes

3. Has the statistical analysis been performed appropriately and rigorously? 

Reviewer #1: Yes

Reviewer #2: Yes

4. Have the authors made all data underlying the findings in their manuscript fully available?

Reviewer #1: Yes

Reviewer #2: Yes

5. Is the manuscript presented in an intelligible fashion and written in standard English?

Reviewer #1: Yes

Reviewer #2: No

6. Review Comments to the Author

Reviewer #1: Thank you to the authors for their engagement with the reviewer comments. They have made several changes and consequently the manuscript is improved. A few minor points remain.

1. There is a discrepancy between the median days to death reported in the abstract, versus that reported in Table 1 / line 283 – this should be corrected.

2. Point 2 from the previous review (“Introduction (paragraph 2): it would be useful to include brief consideration of how important anaemia is clinically in relation to other co-morbidities (i.e. how important is correction of anaemia per se)”).

• The authors have added a sentence referring to reference 1 (Mocroft et al, 1999) stating “Anaemia treatment may even improve survival”. The data presented in the large Mocroft study do not address treatment of anaemia and indeed they conclude (final sentence) that “Further follow-up is needed to determine…. whether an increase in haemoglobin decreases the risk of death”.

• This therefore does not seem an appropriate reference to support the claim – this should be revised or toned down, or appropriate support presented from other sources.

3. Iron deficiency based on MCV:

• I accept that this is now clarified in the recent manuscript. However, I still feel that classifying these individuals as “iron deficient” when the test performance is so poor (AUC=0.54) could be misleading, even though this is the highest (least low) AUC among the markers tested.

• In my opinion, it would therefore be more appropriate to classify explicitly as “low MCV (<83 fL)” or similar (rather than “iron deficiency”), indicating e.g. in the text that this definition is used elsewhere as an index of iron deficiency, but noting that it poorly identified bone marrow iron deficiency in this group (Huibers et al, 2019 – other manuscript).

4. General point: attention needs to be given to manuscript organisation and spelling/grammar in places – a check should be made. For example:

• Remove repetition in discussion: see lines 399-402 and lines 411-414.

• Table S1 / S2: remove repetition in the figure legends (text is duplicated).

• Line 356: sever

• Line 173 and elsewhere: ferritine – change to ferritin

Reviewer #2: Authors have addressed all my concerns. Minor revisions of spelling and grammar recommended in discussion section:

Discussion, line 326: ‘Anaemia in HIV-infected patients is descripted as an independent predictor’ should be corrected to ‘Anaemia in HIV-infected patients is an independent predictor’

Discussion, line 330: ‘reported an estimated one-year mortality of 30 - 55% in severe anaemic vs 3.7%’ should read ‘reported an estimated one-year mortality of 30 - 55% in severely anaemic vs 3.7%’

Discussion, line 344: ‘however our cohort consist with’ should read ‘however our cohort consisted of’

Discussion, line 356: ‘ the complex multifactorial ethology of the severe anaemia might be marker of sever HIV disease’ should read ‘ the complex multifactorial etiology of the severe anaemia might be a marker of severe HIV disease’

Discussion, line 357: 'At last the effect of medication as Zidovudine' should read ‘Lastly, the effects of medications such as Zidovudine’

Discussion, line 358: ‘ However possible due to the bias’ should read ‘however, possibly due to the bias’

Discussion, line 394 and 398- repetition of the statement ‘the prevalence of iron deficiency in our study is higher than…'

Discussion, line 408: 'sever immune suppression' should read 'severe immune suppression’

Discussion, line 413: should read ‘peripheral blood markers such as MCV poorly reflect bone-marrow iron deficiency'

7. PLOS authors have the option to publish the peer review history of their article (what does this mean?). If published, this will include your full peer review and any attached files.

Reviewer #1: No

Reviewer #2: No

---

## [Author Response · Author response to Decision Letter 1]

15 Jan 2020

To: PLOS ONE Kostas Pantopoulos, PhD Academic Editor

Subject: Rebuttal letter resubmission 2 Manuscript ID: PONE-D-19-15827

January 2020

Dear Kostas Pantopoulos PLOS ONE ,

We thank you for accepting our manuscripts titled; “Severe anaemia complicating HIV in Malawi; multiple co-existing aetiologies are associated with high mortality” (PONE-D-19-15827) with minor changes for publication in your journal. We would like to highlight that we initially submitted two manuscripts back to back. The other manuscript titled; “Hepcidin and conventional markers to detect iron deficiency in severely anaemic HIV-infected patients in Malawi” (PONE-D-19-15824) has also been resubmitted with the requested minor changes. Please see below for our responses to the issues raised by the reviewer. We have indicated the original text (with respective line and page numbers) and changes are underlined. As requested we have tracked all changes made in the original manuscript.

It was requested that the raw data should be available as supplementary information for further review, in line with Plos One policy. We are very willing to do so and submitted therefore an anonymised database as a supplementary file. 

Reviewer 1:

1. There is a discrepancy between the median days to death reported in the abstract, versus that reported in Table 1 / line 283 – this should be corrected.

• We have adjusted the abstract. Line 46 (Page2): Overall mortality was high (53%; 100/199) with a median time to death of 17.5 days (IQR 6-55) days.

2. Point 2 from the previous review (“Introduction (paragraph 2): it would be useful to include brief consideration of how important anaemia is clinically in relation to other co-morbidities (i.e. how important is correction of anaemia per se)”).

• The authors have added a sentence referring to reference 1 (Mocroft et al, 1999) stating “Anaemia treatment may even improve survival”. The data presented in the large Mocroft study do not address treatment of anaemia and indeed they conclude (final sentence) that “Further follow-up is needed to determine…. whether an increase in haemoglobin decreases the risk of death”.

• This therefore does not seem an appropriate reference to support the claim – this should be revised or toned down, or appropriate support presented from other sources.

• We agree and have changed the reference: Associations of anemia, treatments for anemia, and survival in patients with human immunodeficiency virus infection. Sullivan : J Infect Dis. 2002 May 15;185 Suppl 2:S138-42.

• Line 76 page 3: Anaemia treatment may even improve survival (7).

3. Iron deficiency based on MCV:

• I accept that this is now clarified in the recent manuscript. However, I still feel that classifying these individuals as “iron deficient” when the test performance is so poor (AUC=0.54) could be misleading, even though this is the highest (least low) AUC among the markers tested.

• In my opinion, it would therefore be more appropriate to classify explicitly as “low MCV (<83 fL)” or similar (rather than “iron deficiency”), indicating e.g. in the text that this definition is used elsewhere as an index of iron deficiency, but noting that it poorly identified bone marrow iron deficiency in this group (Huibers et al, 2019 – other manuscript).

• We appreciate the concerns of the reviewer. Given the use of MCV as marker for iron deficiency in previous reports and the MCV evaluation and explanation given in our sub study, we have taken up the reviewer’s helpful suggestion and have now used MCV ≤ 83 fl as the definition of iron deficiency throughout.

• Lines 242 page 6 of the methods section were adjusted: In a sub study (Huibers et al PLOS One 2019) we evaluated bone marrow (BM) iron deficiency using several conventional blood markers; MCV (fl), MCH (pg/cells), Fe (umol/l), ferritin (ug/dl), TFr1 receptor (nmol/l), TrF index (stFR/Log ferritin). All markers showed suboptimal correlations (i.e. AUCROC < 0.6) with BM iron deficiency, though MCV performed best (AUCROC 0.545). MCV was therefore used to identified iron deficiency as it was the best (though suboptimal) conventional marker in our setting. It was also available for patients beyond the subgroup (n=76) who had a bone marrow result and it is available in most African settings (3, 22, 33). Since an MCV <83fL is commonly used in other studies and guidelines (3, 22, 33) and it was the best predictor of BM iron deficiency in our setting, we used it as the marker for iron deficiency in this analysis. As the AUC-ROC was suboptimal we chose to call it MCV ≤83fL rather than iron deficiency. 

• Iron deficiency was therefore replaced by MCV ≤ 83 fl throughout the analysis/discussion

o Figure 3, Table 2, Table S1 and S2, line 408, 387, 390 (page 15) and line 240 (page 8).

3. General point: attention needs to be given to manuscript organisation and spelling/grammar in places – a check should be made. For example:

• Remove repetition in discussion: see lines 399-402 and lines 411-414.

• Table S1 / S2: remove repetition in the figure legends (text is duplicated).

• Line 356: sever

• Line 173 and elsewhere: ferritine – change to ferritin

• We have adjusted the spelling/grammar accordingly; moreover we have as suggested by the reviewer reviewed spelling and grammar. The sections have been adjusted as requested and the manuscript has been revised by a native English speaker.

Reviewer 2:

4. Discussion, line 326: ‘Anaemia in HIV-infected patients is descripted as an independent predictor’ should be corrected to ‘Anaemia in HIV-infected patients is an independent predictor’

5. Discussion, line 330: ‘reported an estimated one-year mortality of 30 - 55% in severe anaemic vs 3.7%’ should read ‘reported an estimated one-year mortality of 30 - 55% in severely anaemic vs 3.7%’

6. Discussion, line 344: ‘however our cohort consist with’ should read ‘however our cohort consisted of’

7. Discussion, line 356: ‘ the complex multifactorial ethology of the severe anaemia might be marker of sever HIV disease’ should read ‘ the complex multifactorial etiology of the severe anaemia might be a marker of severe HIV disease’

8. Discussion, line 357: 'At last the effect of medication as Zidovudine' should read ‘Lastly, the effects of medications such as Zidovudine’

9. Discussion, line 358: ‘ However possible due to the bias’ should read ‘however, possibly due to the bias’

10. Discussion, line 394 and 398- repetition of the statement ‘the prevalence of iron deficiency in our study is higher than…'

11. Discussion, line 408: 'sever immune suppression' should read 'severe immune suppression’

12. Discussion, line 413: should read ‘peripheral blood markers such as MCV poorly reflect bone-marrow iron deficiency'

• The sections have been adjusted as requested and the manuscript has been revised by a native English speaker. 

We hope our revisions in response to the reviewers’ comments have improved the manuscript and please do not hesitate to contact us if you have any further questions. 

With many thanks for your consideration, on behalf of all the authors.

Minke Huibers, MD, PhD.

---

## [Decision Letter · Decision Letter 2]

23 Jan 2020

Severe anaemia complicating HIV in Malawi; multiple co-existing aetiologies are associated with high mortality

PONE-D-19-15827R2

Dear Dr. Huibers,

We are pleased to inform you that your manuscript has been judged scientifically suitable for publication and will be formally accepted for publication once it complies with all outstanding technical requirements.

With kind regards,

Kostas Pantopoulos, PhD

Academic Editor

PLOS ONE

Additional Editor Comments (optional):

Reviewers' comments:

Reviewer's Responses to Questions

**Comments to the Author**

1. If the authors have adequately addressed your comments raised in a previous round of review and you feel that this manuscript is now acceptable for publication, you may indicate that here to bypass the “Comments to the Author” section, enter your conflict of interest statement in the “Confidential to Editor” section, and submit your "Accept" recommendation.

Reviewer #1: All comments have been addressed

2. Is the manuscript technically sound, and do the data support the conclusions?

Reviewer #1: Yes

3. Has the statistical analysis been performed appropriately and rigorously? 

Reviewer #1: Yes

4. Have the authors made all data underlying the findings in their manuscript fully available?

Reviewer #1: Yes

5. Is the manuscript presented in an intelligible fashion and written in standard English?

Reviewer #1: Yes

6. Review Comments to the Author

Reviewer #1: (No Response)

7. PLOS authors have the option to publish the peer review history of their article (what does this mean?). If published, this will include your full peer review and any attached files.

Reviewer #1: No

---

## [Editor Report · Acceptance letter]

12 Feb 2020

PONE-D-19-15827R2 

Severe anaemia complicating HIV in Malawi; multiple co-existing aetiologies are associated with high mortality 

Dear Dr. Huibers:

I am pleased to inform you that your manuscript has been deemed suitable for publication in PLOS ONE. Congratulations! Your manuscript is now with our production department. 

With kind regards,

on behalf of

Dr. Kostas Pantopoulos 

Academic Editor

PLOS ONE